# Effects of *Spirulina maxima* on a Model of Sexual Dysfunction in Streptozotocin-Induced Diabetic Male Rats

**DOI:** 10.3390/plants12040722

**Published:** 2023-02-06

**Authors:** Eduardo Osel Olvera-Roldán, José Melesio Cristóbal-Luna, Yuliana García-Martínez, María Angélica Mojica-Villegas, Ricardo Pérez-Pastén-Borja, Gabriela Gutiérrez-Salmeán, Salud Pérez-Gutiérrez, Rosa Virginia García-Rodríguez, Eduardo Madrigal-Santillán, José A. Morales-González, Germán Chamorro-Cevallos

**Affiliations:** 1Laboratorio de Toxicología Preclínica, Departamento de Farmacia, Escuela Nacional de Ciencias Biológicas, Instituto Politécnico Nacional, Mexico City C.P. 07738, Mexico; 2Facultad de Ciencias de la Salud/Centro de Investigaciones en Ciencias de la Salud (CICSA), Universidad Anáhuac, Mexico City C.P. 52786, Mexico; 3Departamento de Sistemas Biológicos, Universidad Autónoma Metropolitana-Xochimilco, Mexico City C.P. 04960, Mexico; 4Unidad de Servicios de Apoyo en Resolución Analítica, Universidad Veracruzana, Xalapa C.P. 91190, Mexico; 5Laboratorio de Medicina de Conservación, Escuela Superior de Medicina, Instituto Politécnico Nacional, Mexico City C.P. 11340, Mexico

**Keywords:** diabetes mellitus, sexual behavior, oxidative stress, *Spirulina maxima*

## Abstract

*Arthrospira (Spirulina) maxima* (SM) is a cyanobacterium that has a long history of being used as human food. In recent years, several investigations have shown its beneficial biological effects, among which its antioxidant capacity has been highlighted. The purpose of this study was to evaluate the effects of SM on body weight, glycemia, sexual behavior, sperm quality, testosterone levels, sex organ weights, and the activity of antioxidant enzymes in diabetic male rats (a disease characterized by an increase in reactive oxygen species). The experiment consisted of six groups of sexually expert adult males (n = 6): (1) control (vehicle); (2) streptozotocin (STZ)-65 mg/kg; (3) SM-400 mg/kg; (4) STZ + SM-100 mg/kg; (5) STZ + SM-200 mg/kg; and (6) STZ + SM-400 mg/kg. Sexual behavior tests were performed during the first 3 h of the dark period under dim red illumination. Our results showed that SM significantly improved sexual behavior and sperm quality vs. diabetic animals. Likewise, while the enzymatic activities of SOD and GPx increased, TBARS lipoperoxidation decreased and testosterone levels increased. In view of the findings, it is suggested that SM may potentially be used as a nutraceutical for the treatment of diabetic male sexual dysfunction due to its antioxidant property.

## 1. Introduction

Diabetes mellitus (DM) is characterized by an inability to regulate blood glucose in humans or animals due to either a deficiency of insulin or an erratic signaling pathway [1]. The dysfunction in blood glucose homeostasis affects the metabolism of all macronutrients, not only carbohydrates but lipids and proteins as well [2].

The scientific literature shows that diabetes is on the rise, irrespective of gender [3,4], to the extent that it has reached pandemic proportions [5,6]. DM is more frequent in developing countries due to poor dietary and exercise regimens [7], with 90% of patients presenting type 2 DM [8].

DM is also a risk factor for secondary disorders, such as coronary disease, stroke, chronic kidney disease, loss of vision, neuropathies, and sexual disorders [9]. Regarding sexual disorders, DM is closely associated with sexual dysfunction, leading to infertility due to alterations in spermatogenesis, structural changes in testicular tissue, the alteration of glucose metabolism in Sertoli cells, a decrease in testosterone concentrations, ejaculatory alteration, and a decrease in libido [7]. In fact, the incidence of erectile dysfunction in diabetic patients has been reported at between 35–75%, i.e., three times higher than that of non-diabetic men [10].

Although a sole isolated molecular pathway has not been identified as the cause of sexual dysfunction in DM, one of the suggested mechanisms is oxidative stress (OS) due to an overproduction of the reactive oxygen species (ROS) concomitant with a decrease in endogenous antioxidant activity [11].

There are certainly some synthetic drugs for improving sexual performance in men with DM; nevertheless, their high cost and the potential for adverse effects have engendered interest in natural products that are more economical and exhibit a safer profile while alleviating sexual dysfunction [12,13].

*Arthrospira maxima* (*Spirulina maxima*) (SM) is a filamentous cyanobacterium (previously considered blue-green algae) that has been cultivated and consumed in many parts of the world since ancestral times [14]. SM possesses a widely appreciated nutritional value and pharmacological effects, and attention has been drawn for some years both by in vivo and in vitro experiments [15,16,17] to its antioxidant activity, which is mainly due to its high content of antioxidant compounds, such as phycobiliproteins, beta-carotene, tocopherols, and phenolic acids [18]. In this sense, SM offers many functional bioactive ingredients with anti-inflammatory, antimetastatic, immunostimulatory, cardioprotective, and metalloprotective activity [19,20,21,22], and it has been effective in the treatment of neuropathies [23] and neurobehavioral and cognitive deficits [24,25]. Furthermore, it has been determined that the SM dose-dependently activates cellular antioxidant enzymes and inhibits peroxidation and DNA damage. In the same way, phycobiliproteins of SM, such as phycocyanin, have a broad capacity to capture free radicals, such as hydroxyl, alkoxyl, and peroxyl radicals, to inhibit liver microsome lipid peroxidation and increase the activity of the antioxidant enzymes superoxide dismutase (SOD) and catalase (CAT) during the process of oxidative stress [19,20]. However, to date, no study has, to our knowledge, been carried out to evaluate the benefits of the antioxidant effect of SM on the alterations generated in sexual behavior induced by the oxidative stress characteristic of diabetes.

Given the previously mentioned information, this study aimed to evaluate the potential effects of SM in restoring impaired sexual behavior, improving spermatic parameters, reducing oxidant effects in testicular tissue, increasing antioxidant enzyme activity, and restoring testosterone levels in streptozotocin/nicotinamide-induced diabetic male mice.

## 2. Results

### 2.1. Body Weight and Glycemia

Final body weights are presented in Figure 1. As observed, STZ, STZ + SM 100, and STZ + SM 200 were significantly lower compared with the control group, whereas this was not found for STZ + SM 400. Only SM 400 differed from STZ.

Glycemia is presented in Figure 2. Both the control and SM 400 groups demonstrated normal concentrations of ~90 mg/dL, whereas the remainder presented a significant increase in such values, reaching more than 300 mg/dL in STZ and STZ + SM 100. Hyperglycemia was, however, attenuated but significantly different compared with that of the control group when SM was administered at doses of 200 and 400 mg/kg.

### 2.2. Copulatory Behavior

As depicted in Figure 3, mount latency (ML), intromission latency (IL), ejaculatory latency (EL), and ejaculatory series duration (ESD) were significantly longer in the STZ groups compared with the control group; contrariwise, mount frequency (MF) and intromission frequency (IF) were significantly lower, thus evidencing sexual dysfunction.

Compared with the STZ group, a significant difference (*p* < 0.05) was found for SM groups in mount latency (ML) for SM 400, STZ + SM 200, and STZ + 400, as well as in intromission latency (IL) for the SM 400 and STZ + SM 400 groups and ejaculatory latency (EL) with the treatments of SM 400, STZ + SM 200, and STZ + SM 400.

### 2.3. Sperm Quality

Table 1 presents the results of the spermatic quality analysis; it can be appreciated that sperm count, motility, and viability were significantly (*p* < 0.05) inhibited in diabetic rats. On the other hand, sperm abnormalities were markedly (*p* < 0.05) increased in this group.

All animals treated with SM exhibited a significantly (*p* < 0.05) higher total sperm count than those in the STZ group. Sperm motility was found to be diminished in the STZ + SM 100 and STZ + SM 400 groups compared with the control group, but it increased significantly (*p*< 0.05) in STZ + SM 200 compared with diabetic animals.

The viability analysis revealed a significant increase in all SM-treated groups, except for the STZ + SM 100 dose, which, notwithstanding this, exhibited a positive trend. Finally, the morphology of the sperm in SM-treated rats showed a significantly decreased percentage of abnormalities compared with that of diabetic animals.

### 2.4. Sex Organ Weight

No differences were found in the weight of seminal vesicles, testes, and epididymis among the groups (data not presented).

### 2.5. Biochemical Analyses

According to Table 2, a significant increase (*p* < 0.05) in lipoperoxidation was found in the diabetic groups, together with a decrease in SOD and GPX. However, treatment with SM significantly (*p* < 0.05) improved the activity of these antioxidant enzymes compared with that of the diabetic animals. 

### 2.6. Testosterone Analysis

The results of the analysis of the serum testosterone concentration are presented in Figure 4. In the STZ group, a significant decrease (*p* < 0.05) in the concentration with regard to the control was manifested. However, in the group treated with SM and the diabetic groups treated with SM, the concentration of this hormone increased significantly (*p* < 0.05) in the diabetic group and was very similar to that in the control group.

## 3. Discussion

Sexual dysfunction is one of the most frequent complications in persons with diabetes for which, to our knowledge, there is no effective treatment to date [26], although some hypoglycemic agents may be useful in the treatment of diabetes [27]. Numerous authors have shown that male sexual dysfunction derived from diabetes is associated with oxidative stress and, consequently, a decrease in antioxidant concentrations in diabetic patients [28]. The said imbalance of prooxidant and antioxidant species gives rise to the production of free radicals [29], the increase in the destruction of nitric oxide (NO), and the increase in the peroxidation of polyunsaturated fatty acids (PUFA) [30]. For these reasons, we hypothesized that a natural product with great antioxidant activity, such as SM, when administered to diabetic male rats, would reduce or prevent damage to sexual behavior and other biochemical parameters related to it.

Almost all the components of SM, in addition to the pharmacological effects mentioned above and the absence of toxicity, have been effective in the treatment of neuropathies [23] and neurobehavioral and cognitive deficits [24,25]. In the present work, we tried to improve sexual behavior, blood and serum biochemical parameters and hormones, and antioxidant capacity by assessing the serum and testis MDA and Nrf2 pathway of reproductive organs of diabetic male rats by feeding the animals with SM [12]. No previous evidence, to our knowledge, has been reported on the effects of this algae in sexual dysfunction models, particularly assessing its antioxidant potential in diabetes models.

We found that, except for PEI, in which there was only an increasing trend in diabetic animals compared with controls, sexual behavior indicators significantly increased in ML, IL, and EL as well as in EDS, but these decreased in MF and IF, all of which is in agreement with the results of Al-Oanzi [29] and De [27], demonstrating deficiencies in the carrying out of copulation in diseased animals.

A significant amelioration in these disrupted parameters was observed in animals treated with SM. Thus, a decrease in ML, EL, and IL compared with the diabetic group indicates an increase in sexual motivation [31,32]. On the other hand, the significant increase in MF, as in the case of STZ + SM + 400, and the tendency of the increase in IF in the case of STZ + SM 200 and STZ + 400 reveal that diabetes did not interfere with the activity of cyanobacteria for sexual provocation in terms of the efficacy of raising penile posture and consequently sexual vigor, libido, and potency, characteristics demonstrating that the treated rats were aroused [33]. Likewise, because sexual dysfunction in diabetes is associated with inflammation [34], it is probable that the anti-inflammatory activity of SM contributed to these results. The phycobiliproteins, β-carotene and other vitamins, chlorophyll, and phenols present in SM may have been responsible for sexual enhancement due to their antioxidant properties, which could be the subject of future studies.

Regarding the endocrine profile, the decrease in the serum testosterone level in male animals to which STZ was administered for the induction of diabetes was consistent with the results previously published by various authors [35,36], which showed that testosterone is necessary for normal development of male sexual behavior [28]. SM increased this level and reached values similar to those of the control group, which, similar to other agents, influenced erectile function, including sexual desire, until penile erection [30]. In this regard, it is noteworthy that in other studies, SM increased testosterone concentrations in the testes when it was administered to rats that had decreased concentrations of this hormone due to the administration of toxic agents, such as lead acetate [37], and when it was administered to mice via the administration of bifenthrin [38]. SM contains small concentrations of androgenic molecules, such as cholesterol and fatty acids, among other components that can influence testosterone production [33], as do quercetin, vitamin E, and vitamin C [39,40]. Moreover, it is thought that normality in the serum testosterone concentration may be more related to the cytoprotective effect of SM in the testicular tissue, specifically in Leydig cells. On the other hand, there are a few discrepancies in the influence of testosterone concentration on some parameters of sexual activity in rats [41].

In addition to the relationship between the improvement of sexual behavior and the concentration of testosterone, the reduction in oxidative stress caused by SM is probably a decisive factor in the pathophysiology of sexual dysfunction in that it is associated with an overproduction of free radicals [29]. In our work, which found a decrease in TBARS and an increase in the enzymatic activities of SOD and GPX in the testicular tissue of rats treated with SM, enzymes were considered to be the main antioxidants responsible for maintaining the optimal concentration of ROS [42]. Is important to note that, although SOD activity is typically assumed to be within cells, there is also an extracellular isoform—described in 1982—that differs from the cytoplasmatic SOD, as it is conformed as a tetramer (rather than a homodimer) and contains six Cys residues (vs. four in the intracellular). In this sense, both SODs catalyze the conversion of O_2_^−^ into H_2_O_2_, which means that the changes in its activity have an impact not only on maintaining the balance between the prooxidant and antioxidant species but also decreasing blood glucose levels. Under hyperglycemic conditions, endothelial cells increase the levels of O_2_^−^; such an overproduction of O_2_^−^ inhibits the activity of glyceraldehyde-3-phosphate dehydrogenase (GAPDH), a very important enzyme in the glycolytic pathway, and the inhibition of GAPDH leads to the accumulation of glucose and other intermediate metabolites of this pathway and shifts to other alternative pathways of glucose metabolism [43].

Regarding sperm parameters, diabetic rats exhibited lower sperm quality compared with the control group, presenting a lower sperm count, lower motility and viability, and an increase in the percentage of abnormalities; this again demonstrates the effects of oxidative stress and the production of free radicals, such as ROS, caused by hyperglycemia [35] due to the peroxidation of polyunsaturated fatty acids (PUFA) in sperm cell membrane spermatozoa [30]. Treatment with SM significantly increased the value of the previously mentioned parameters, which reveals its spermatic efficacy due to its antioxidant activity, although a dose–response relationship was not always observed. This antioxidant activity of SM has been the mechanism for explaining many of its pharmacological activities demonstrated in vivo and in vitro [44].

There were interesting results in this study, but prior to the extrapolation to humans mimicking the diabetic conditions of both species, it is necessary to better understand why SM improves sexual dysfunction parameters.

## 4. Materials and Methods

SM was kindly donated by Alimentos Esenciales para la Humanidad, S.A. de C.V. (Mexico City, Mexico) as a fine powder with a green-blue appearance. It was stored in trilaminate bags of metalized polyester to avoid light and air exposure and was maintained at room temperature until its use. In previous studies carried out in our laboratory, it was determined that the chemical and biochemical composition of this product for each 100 g was as follows: loss on drying (humidity) 4.65 g, lipids <0.50 g, saturated fats <0.10 g, crude fiber 0.96 g, proteins 60.08 g, ashes 6.71 g, total carbohydrates 27.60 g, sodium 114.99 mg, and energy supply 350 kcal. The microbiological analysis showed the following: fecal coliforms <3 NMP/100 mL, total coliforms <3 NMP/100 mL, *Escherichia coli* absent, mushrooms <1 CFU/g, yeasts <1 CFU/g, aerobic mesophilic 39,000 CFU/g, *Salmonella* absent, and *Staphylococcus aureus* absent. In addition, the metal analysis showed the following: As 0.054 mg/g, Al 0.352 mg/g, Cu <0.016 mg/g, Ca 1.1 mg/g, Fe 0.513 mg/g, Na 10.1 mg/g, K 12.6 mg/g, Cd <0.016 mg/g, Mn 0.030 mg/g, Ni <0.033 mg/g, Pb <0.033 mg/g, Mg 2.7 mg/g, Zn 0.016 mg/g, and Hg 0.0001 mg/g [14].

### 4.1. Animals

Thirty-six Wistar albino male rats of 350 ± 15 g (*n* = 6) and ten female Wistar albino rats 250 ± 15 g of eighty weeks old were obtained from the breeding colony of the Universidad Autónoma Metropolitana (UAM), Unidad Xochimilco, Mexico City, Mexico. They were housed in polypropylene cages with a sawdust floor and placed in an air-conditioned room (22–23 °C; 50–60 % humidity; and artificial illumination with an inverted light-dark cycle of 12h/12 h; lights on at 7:00 pm). The rats had access to standard rodent chow and purified water ad libitum, and they were adapted to the laboratory environment for 15 days prior to beginning the experimental protocols.

All procedures, including euthanasia, were performed in agreement with the Bioethics Committee of the National School of Biological Sciences of the National Polytechnic Institute, Mexico City ZOO-021-2019 and following the Mexican Official Regulation (NOM ZOO-062-200-199) entitled “Technical Specifications for Production, Care, and Use of Laboratory Animals”.

### 4.2. Preparation of Females for Couplings, Animal Selection, and Preparation for Experiments

With the purpose of carrying out the definitive study of sexual behavior with sexually expert males, they were previously trained in mating with receptive females in circular acrylic cages 60 cm in diameter by 40 cm high on a bed of sawdust on alternate days to avoid sexual exhaustion. Only sexually apt healthy males were selected [45]. Only males who completed the copulatory sequence in <15 min from the entry of the female into the cages in three consecutive copulation sessions were considered sexually apt. Animals (males and females) that did not complete the copulatory sequence during training in the required time were removed from the study and replaced with other animals [45]. Receptive females were used for couplings, for which a bilateral ovariectomy was previously performed on female rats, as described by Khajuria [46]. Female rats were anesthetized with an i.p. dose of sodium pentobarbital, 60 mg/kg, and under the effect of anesthesia, both ovaries were excised, and the oviducts were ligated. The animals were maintained in recovery for 15 days after surgery. Finally, before initiating the sexual-behavior experiment, estrus was further induced in these ovariectomized rats by subcutaneous (s.c.) administration of estradiol benzoate (12 µg/kg) (Sigma Chemical Co., Ltd., St. Louis, MO, USA) 24 h before starting sexual behavior experiments and progesterone (3 mg/kg) (Sigma Chemical Co., Ltd.) 4 h before starting sexual experiments.

### 4.3. Induction of Diabetes Type II in Males

Once the sexually expert males were obtained, diabetes was induced (except for animals in the normal control group and SM 400 mg/kg group) with a single dose of STZ 65 mg/kg i.p. dissolved in citrate buffer at pH 4.6. One hour after STZ administration, 110 mg/kg of nicotinamide dissolved in saline solution was administered i.p., thus giving rise to the development of type II diabetes, according to the method described by Masiello [47]. After diabetes induction, blood glucose concentration was measured weekly from the dorsal tail vein by puncture under overnight fasting conditions using a glucometer (Accu-Chek Performa; Roche, Germany). Rats were considered diabetic when the blood glucose concentration was >200 mg/dL.

### 4.4. Mating Behavior Test

The sexually expert rat males (in whom diabetes was induced) were randomly distributed into six groups (*n* = 6), and the following treatments were administered for 5 weeks: 1. Control; 2. streptozotocin (STZ) 65 mg/kg; 3. SM 400 mg/kg; 4. STZ + SM 100 mg/kg; 5. STZ + SM 200 mg/kg, and 6. STZ + SM 400 mg/kg. The rats were then evaluated with sexual behavior tests. SM was suspended in water with 1% Tween-80 and administered orally (per os) in a constant volume per kg of weight. The doses of SM used in this study (100, 200, and 400 mg/kg) were selected on the basis of the results of pharmacological studies in rats carried out by our working group; such doses have shown good efficacy without producing mortality or altering body weights, tissues, and organs [14,48]. The evaluation in sexual behavior tests was performed during the first 3 h of the dark period under dim red illumination in a silent room, as previously described by Fumero [49]. Each male was introduced into a cage where it was first acclimatized for 10 min, then the female was introduced, and the registrations of the following parameters were videotaped with a digital camera (Sony HDR-CX405; China): mount latency (ML), counted as the time elapsed since the introduction of the female into the male box until the first mount; intromission latency (IL), calculated as time from introduction of the female to the first intromission by the male rat; mount frequency (MF), defined as the number of mounts before ejaculation; intromission frequency (IF), considered the number of intromissions before ejaculation; ejaculation latency (EL), recorded as the time elapsed from the first intromission to the first ejaculation; post-ejaculatory interval (PEI), calculated by the time from the first ejaculation to the next intromission by the male rat; and ejaculatory series duration (ESD), the time elapsed from the introduction of the female to the beginning of the second ejaculatory series (marked by the next intromission after the post-ejaculatory interval) [30,40].

Once tests were completed, the males were sacrificed by cervical decapitation. Blood was collected from the descending aorta vein, further incubated at room temperature for 10 min to allow coagulation, and subsequently centrifuged at 5000 rpm for 5 min to obtain the serum, which was frozen at −70 °C until analysis.

### 4.5. Sperm Collection and Analysis

Subsequently, the males’ sacrificed testes, epididymis, and seminal vesicles were excised and weighed. A sperm sample was obtained from the vas deferens and the tail of the epididymis, which was suspended in 1 mL of Hanks’ balanced salt solution preheated to 37 °C.

Staining was performed by incubating 10 µL of sperm sample with 10 µL of eosin–nigrosin vital dye at 36 °C for 5 min; this then was spread and dried on a slide to observe the viability of the cells under a microscope with a 100X objective. Viable (unstained) cells were distinguished from those that were not (stained), and the percentage of each was yielded from a count of at least 200 cells. Their morphology was also evaluated without characterization of the abnormality types found.

### 4.6. Biochemical Analysis

The determination of enzymatic activity in the testes was performed using commercial kits for superoxide dismutase (SOD) (Ransod; Randox Laboratories, Ltd., Crumlin, UK), which was based on the method described by Andersen [50], and glutathione peroxidase (GPX) (Ransel; Randox Laboratories, Ltd., Crumlin, UK) was determined according to the method of Paglia [51]. Lipoperoxidation was assessed by malondialdehyde (MDA), which was determined by the TBARS technique described by Matsuzawa [52]. 

Testosterone in the serum was determined with an ELISA kit (Cayman Chemical Company, Ann Arbor, MI, USA) following the manufacturer’s instructions.

### 4.7. Statistical Analysis

The parametric data (body weight, blood glucose levels, mount latency, intromission latency, ejaculation latency, post-ejaculatory interval, and ejaculatory series duration; the count, motility, viability, and abnormality of the sperm; TBARS levels, SOD and GPX concentration; and testosterone levels) were expressed as the mean ± SEM (*n* = 6), while non-parametric data (mount and intromission frequencies) were expressed as frequencies. In the first case, after a normality test, analysis comparisons between multiple groups were performed with one-way ANOVA and post hoc Dunnett tests; in turn, for non-parametric data, Kruskal–Wallis with Dunnett post hoc tests were carried out [53]. In both cases, significant differences were considered when *p* < 0.05. All statistical analysis and the preparation of figures were made on GraphPad Prism v.8.

## 5. Conclusions

Oxidative stress in diabetic rats impairs sexual behavior in males, modifying mainly spermatic and biochemical parameters and testosterone levels. The antioxidant effect of SM, evidenced by the increase in the activity of SOD and GPX and the decrease in MDA in male diabetic rats, effectively counteracted the deterioration in parameters such as sexual conduct, sperm count, sperm motility, sperm viability, and sperm abnormalities and preserved sexual conduct together with higher sperm quality. These results show that antioxidant therapy, specifically with SM, could have a beneficial application in the treatment of diabetic sexual disorders in men.

## Figures and Tables

**Figure 1 plants-12-00722-f001:**
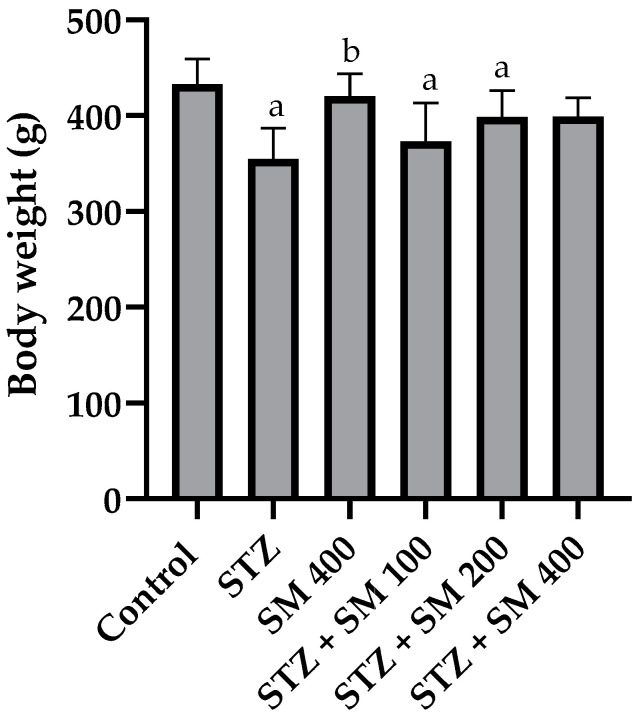
Effect of SM on body weight (g). STZ, streptozotocin; SM, *Spirulina maxima*. Each bar indicates the mean ± SEM (*n* = 6). Significant difference (*p* < 0.05) vs. ^a^ control; ^b^ STZ 65 mg/kg.

**Figure 2 plants-12-00722-f002:**
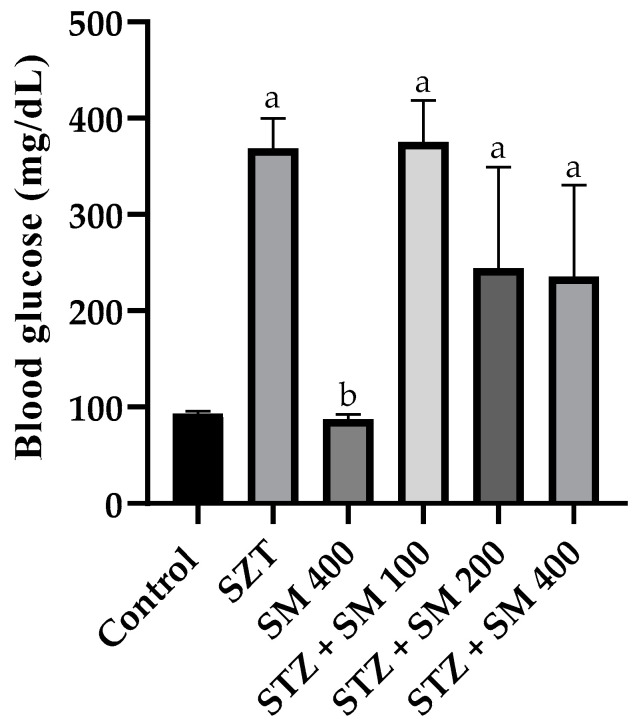
Effect of SM on blood glucose levels. STZ, streptozotocin; SM, *Spirulina maxima*. Each bar indicates the mean ± SEM (*n* = 6). Significant difference (*p* < 0.05) vs. ^a^ control; ^b^ STZ 65 mg/kg.

**Figure 3 plants-12-00722-f003:**
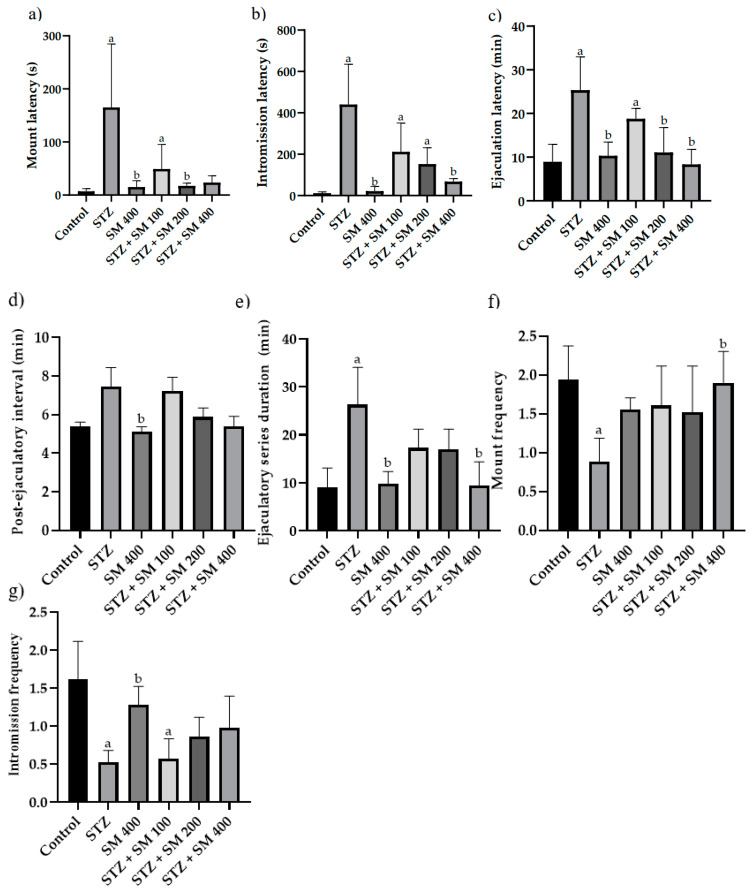
Effect of SM on copulatory behavior. (**a**) Mount latency; (**b**) intromission latency; (**c**) ejaculation latency; (**d**) post-ejaculatory interval; (**e**) ejaculatory series duration; (**f**) mount frequency; (**g**) intromission frequency. STZ, streptozotocin; SM, *Spirulina maxima* (SM). Each bar indicates the mean ± SEM (*n*= 6). Significant difference (*p* < 0.05) vs. ^a^ control; ^b^ STZ 65 mg/kg.

**Figure 4 plants-12-00722-f004:**
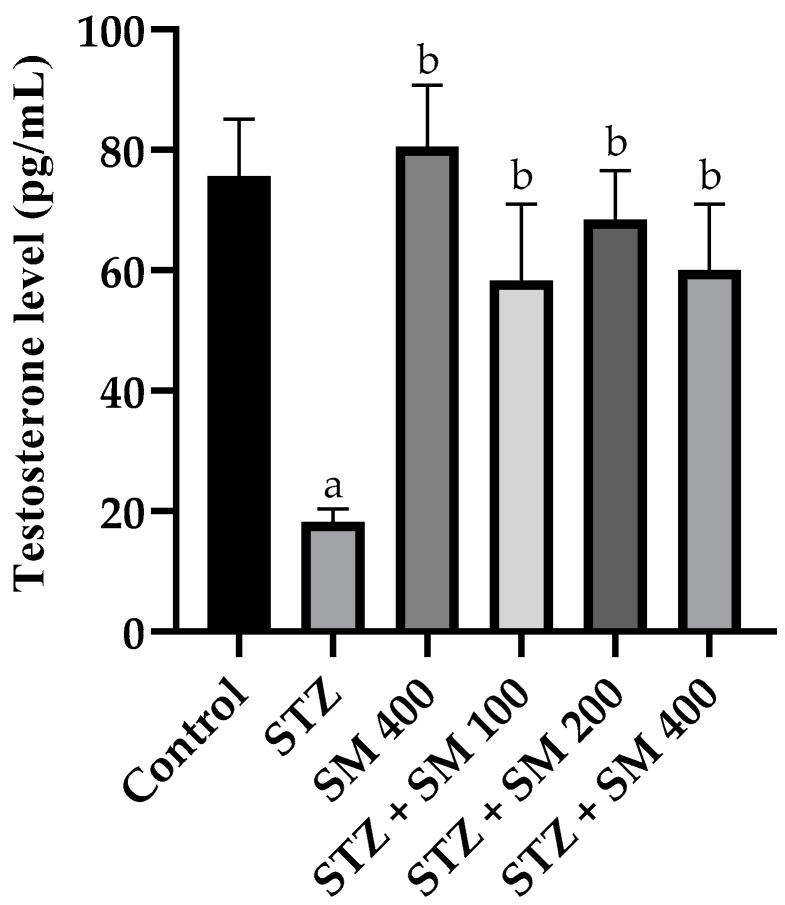
Effect of SM on testosterone levels. STZ, streptozotocin; SM, *Spirulina maxima*. Each bar indicates the mean ± SEM (*n* = 6). Significant difference (*p* < 0.05) vs. ^a^ control; ^b^ STZ 65 mg/kg.

**Table 1 plants-12-00722-t001:** Effect of SM on count, motility, viability, and abnormality of the sperm of control and experimental groups.

Treatment (mg/kg)	Sperm Count (X106/mL)	Sperm Motility	Sperm Viability	Sperm Abnormality (%)
Control	69.42 ± 2.72	29.77 ± 7.36	61.74 ± 1.20	3.93 ± 0.98
STZ	39.44 ± 6.05 ^a^	9.48 ± 2.58 ^a^	27.56 ± 2.94 ^a^	6.08 ± 1.17 ^a^
SM 400	75.29 ± 6.33 ^b^	20.52 ± 2.99 ^b^	61.49+ 2.20 ^b^	4.3 ± 1.13 ^b^
STZ + SM 100	64.04 ± 6.22 ^b^	7.55 ± 1.89 ^a^	33.45 ± 4.00	4.1 ± 1.24 ^b^
STZ + SM 200	72.50 ± 2.81 ^b^	16.38 ± 2.87 ^b^	44.95 ± 2.68 ^b^	3.9 ± 1.20 ^b^
STZ + SM 400	74.20 ± 2.95 ^b^	8.52 ± 1.52 ^a^	43.82 ± 3.86 ^b^	3.2 ± 1.12 ^b^

The animals were given the corresponding treatment for 5 weeks. STZ, streptozotocin; SM, *Spirulina maxima*. Results are expressed as the mean ± SEM (*n* = 6). Significant difference (*p* < 0.05) vs. ^a^ control; ^b^ STZ 65 mg/kg.

**Table 2 plants-12-00722-t002:** Effect of SM on serum biochemical parameters in male diabetic rats.

Treatments (mg/kg)	TBARS (nmol/mg Protein)	SOD (U/mg Protein)	GPX (mU/mL)
Control	12.57 ± 0.42	4.11 ± 0.03	436.20 ± 24.63
STZ	17.65 ± 1.25 ^a^	1.59 ± 0.04 ^a^	209.83 ± 1.83 ^a^
SM 400	13.91 ± 0.69 ^b^	4.24 ± 0.02 ^b^	430.10 ± 25.17 ^b^
STZ + SM 100	13.79 ± 0.45 ^b^	2.81 ± 0.06 ^b^	332.65 ± 10.75
STZ + SM 200	11.86 ± 0.64 ^b^	4.28 ± 0.03 ^b^	451.55 ± 11.55 ^b^
STZ + SM 400	12.38 ± 0.79 ^b^	4.27 ± 0.02 ^b^	447.42 ± 11.72 ^b^

The animals were given the corresponding treatment for 5 weeks. STZ, streptozotocin; SM, *Spirulina maxima*. Results are expressed as the mean ± SEM (*n*= 6). Significant difference (*p*< 0.05) vs. ^a^ control; ^b^ STZ 65 mg/kg.

## Data Availability

Not applicable.

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
