# Peer review of "Effects of Spirulina maxima on a Model of Sexual Dysfunction in Streptozotocin-Induced Diabetic Male Rats"

_plants, 2023, doi:10.3390/plants12040722_

Round 1

Reviewer 1 Report

Thanks for the opportunity to review this research. The manuscript entitled Effects of Spirulina maxima on a model of sexual dysfunction in streptozotocin-induced diabetic male ratsis interesting but contain a lot of mistakes and is not explained in a correct way. The discussion is very incomplete and there is a lot of section that must rewrite or improved.

Comments:

Abstract: the methodology is not well described in the abstract.

Introduction: more information was needed about the antioxidant activity.

Material and methods: more information was needed about the statistical analysis and the references of the software used. The section Materials and Methods must be rewrite is to confuse, incomplete and in some parts not very well described.

Results and discussion: the discussion is very poured and there aren’t conclusions. The problem of a missing hypothesis is also reflected in the results and discussion part of the manuscript.

Please improve the conclusion of the manuscript and check the text for technical errors. I recommend the publishing of the paper after the necessary corrections.

Author Response

Responses to Reviewer 1

Comment 1: Abstract: the methodology is not well described in the abstract.

Response 1: Thank you for your observation. Since the format of the Plants requires an extension of no more than 200 words in the abstract. It was very complicated for us, to describe the methodology in more detail in such extension. However, the abstract was restructured to make the methodology clearer (lines 18-27).

Comment 2: Introduction: more information was needed about the antioxidant activity.

Response 2: Thank you for your insightful observations. After analyzing your recommendation, we agree that the adjustments you propose will make the context clearer. Therefore, more information was added regarding the antioxidant activity of SM (lines 62-74).

Comment 3: Material and methods: more information was needed about the statistical analysis and the references of the software used. The section Materials and Methods must be rewrite is to confuse, incomplete and in some parts not very well described.

Response 3: Thank you for pointing out that situation. The section Materials and Methods has been corrected and restructured with the hope that it is now clearer (lines 280-399).

Comment 4: Results and discussion: the discussion is very poured and there aren’t conclusions. The problem of a missing hypothesis is also reflected in the results and discussion part of the manuscript.

Response 4: Thank you for your observation. The sections Discussion have been corrected and restructured with the hope that it is now clearer (lines 194-262).

Comment 5: Please improve the conclusion of the manuscript and check the text for technical errors. I recommend the publishing of the paper after the necessary corrections.

Response 5: Thanks for your comments; the manuscript was reviewed in search of technical details, which were addressed in the corresponding sections. Also, the discussion section was improved (lines 401-405).

Reviewer 2 Report

This well done study presents the effects of Spirulina maxima (SM) on the body weight, glycemia, sexual behavior, sperm quality, testosterone level, and some parameters of the redox status of plasma in rats with streptozotocin-induced diabetes. While the effect of SM on the glycemia was devoid of statistical significance, SM beneficially affected sexual behavior, sperm count, viability and motility, and reduced sperm abnormality. In the blood serum, SM administration increased the testosterone level reduced the TBARS level and increased the activities of superoxide dismutase and glutathione peroxidase. The results are unequivocal and point to beneficial action on the sexual functions of diabetic rats.

Remarks

The preparation of SM requires some characterization; the information that it was a donated powder of green-blue appearance is not sufficient.

Superoxide dismutase is mainly an intracellular enzyme. How do the authors interpret changes in SOD activity in blood serum?

Lines 198/199: How can SOD present in SM given per os can exert antioxidant effect in the rat?

Author Response

Responses to reviewer 2

Comment 1: This well done study presents the effects of Spirulina maxima (SM) on the body weight, glycemia, sexual behavior, sperm quality, testosterone level, and some parameters of the redox status of plasma in rats with streptozotocin-induced diabetes. While the effect of SM on the glycemia was devoid of statistical significance, SM beneficially affected sexual behavior, sperm count, viability and motility, and reduced sperm abnormality. In the blood serum, SM administration increased the testosterone level reduced the TBARS level and increased the activities of superoxide dismutase and glutathione peroxidase. The results are unequivocal and point to beneficial action on the sexual functions of diabetic rats.

Response 1: Thank you for your kind comments.

Comment 2: The preparation of SM requires some characterization; the information that it was a donated powder of green-blue appearance is not sufficient.

Response 2: Thank you for your observation. In previous publications, the Spirulina that the company “Alimentos Esenciales para la Humanidad, S.A. de C.V.” donates to us has been characterized. Therefore, a summary of the most important elements of said analysis has been added to the manuscript (lines 280-290).

Comment 3: Superoxide dismutase is mainly an intracellular enzyme. How do the authors interpret changes in SOD activity in blood serum?

Response 3: Thank you for your question. Indeed, SOD activity is typically assumed to be within cells; however, there is also an extracellular isoform -described in 1982- which differs from the cytoplasmatic enzyme as it is conformed as a tetramer (rather than a homodimer) and contains 6 Cys residues (vs. 4 in the intracellular). The study by Boriskin (IOP Conf Ser Earth Environ 2019; 403) demonstrated a statistically significant correlation between the two isoforms, both intracellular and plasmatic SOD. In this sense, the activity of SOD is determinant for the control not only of oxidative stress in diabetic patients, but also in glucose bitransformation. The complete information is described in more detail in lines 251-261.

Comment 4: Lines 198/199: How can SOD present in SM given per os can exert antioxidant effect in the rat?

Response 4: Thank you for the observation. We agree; although SOD exerts its antioxidant activity within SM’s cells it is not significantly absorbed in the rat. We have hence eliminated this idea (lines 226-228).